# Distribution Pattern and Factors Influencing Spontaneous Plant Diversity in Different Wetland Habitats

Yifan Yang [1,†], Bin Xu [1,†], Qingqing Yu [2], Likun Fan [3], Tingting Guo [1], Dongshi Fu [1], Hao Chen [1], Hai Yan [1], Feng Shao [1,*] and Xiaopeng Li [4,*]

1. School of Landscape Architecture, Zhejiang Agriculture and Forestry University, Hangzhou 311300, China
2. School of Architecture, China Academy of Art, Hangzhou 310024, China
3. Qianjiang Management Office, Hangzhou West Lake Scenic Area Management Committee, Hangzhou 310008, China
4. School of Architecture, Southwest Jiaotong University, Chengdu 611756, China
* Correspondence: shaofeng@zafu.edu.cn (F.S.); penguinlee26@swjtu.edu.cn (X.L.); Tel.: +86-1345-682-9121 (F.S.); +86-1381-167-1315 (X.L.)
† These authors contributed equally to this work.

**Abstract:** Wetlands contain a large number of spontaneous plants, and the ecological value of such plants should not be underestimated. However, the influence of the surrounding environment on the composition of spontaneous plants in wetlands is still unclear. Hangzhou Jiangyangfan Ecological Park, built more than 20 years ago, is the first ecological park in China based on the "wild state" concept. The wetland in the park was taken as a study case, and we investigated some of the ecological factors affecting the diversity and distribution pattern of spontaneous plants in wetland habitats after natural succession. A total of 100 species of spontaneous plants were recorded, belonging to 93 genera and 48 families, with native species accounting for approximately 78% of the total. We found significant differences in the species diversity and distribution patterns of spontaneous plants in different habitats and microhabitats. According to the biological characteristics of spontaneous plants, the fruit types were mostly achenes and capsules, and the seed dispersal mode was mainly animal dispersal. Different fruit types and dispersal modes affected the composition and distribution of spontaneous plants. In terms of environmental factors, the water depth and slope aspect were the key factors determining the diversity and distribution pattern of spontaneous plants. It was also found that the clustering degree of cultivated plants had an effect on the composition of spontaneous plants. To form a more natural wetland landscape, it is necessary to provide a variety of growing environments for spontaneous plants. We suggest allocating appropriate habitat types in wetlands and reducing human intervention to increase biodiversity.

**Keywords:** spontaneous plants; species diversity; distribution pattern; seed dispersal; environmental heterogeneity; influencing factor

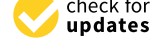



## 1. Introduction

In recent years, to improve the urban environment, ideas and strategies such as "urban rewilding" and "natural-based solution" have been increasingly put forwards. Vegetation plays an important ecological role in the city. In urban environments with different substrates, the composition of plant communities will change due to historical factors such as land use and management patterns [1]. Urbanization does not necessarily lead to large-scale vegetation degradation, but it negatively affects the ecological and physical environment [2]. The improvement of the ecological environment can increase the richness of biodiversity by enhancing the complexity and heterogeneity of vegetation [3]. The physical environment can be improved by increasing the urban vegetation cover. On the one hand, it can help to reduce the surface temperature and alleviate the urban heat island effect [4,5]. On the other hand, it can indirectly improve ecosystem service functions

and promote the urban carbon cycle [6]. In recent years, green construction projects have improved the overall urban vegetation cover rate to a certain extent [7]. However, there is a threshold for the cooling and humidifying effect of urban plant communities [8]. It is suggested that we should not only pursue vegetation coverage, but also adopt a better vegetation allocation scheme.

In an open environment such as a garden, people have different preferences for plant selection in plant communities [9]. In a relatively closed environment, high greenery quantity and flowering plants are the most preferred [10]. In the actual construction process, people tend to grow plants with higher aesthetic and economic value. Although people's choice and application of plants are diversified, there are still studies showing that the flora in China is facing the risk of biological homogenization. Due to human activities, narrowly distributed species are being replaced by widely distributed species, and native species are being replaced by non-native species [11]. Additionally, the passage of time increases community similarity [12]. In a study of vegetation in Potchefstroom, it was found that although there was no homogenization phenomenon at present, the richness of local herbaceous plants had decreased [13]. The fact that homogenization had not yet occurred may be related to human management and other factors, because high human disturbance will increase alien species [14]. Compared with exotic plants, native plants are more beneficial to the stability of plant communities and the protection of plant diversity in the region.

Wild vegetation often grows in urban wastelands. In most cases, urban wastelands have a higher species richness [15] and a higher proportion of native species than other urban green spaces. An increasing number of scholars have paid attention to the protection of wild vegetation and the ecological potential of improving urban biodiversity. Weeds can be found everywhere in cities, such as on roofs [16], walls [17], parks [18] and ponds [19]. In the 1970s, the term "spontaneous vegetation" was used by -ecologists to refer to plants that naturally settled and grew, which was different from the meaning of "weeds". Since then, research in China and abroad has been carried out using the term "spontaneous plants". Some Chinese scholars conducted an investigation on the spontaneous plants in Chongqing, and the results showed that most of the spontaneous plants in the city were native species, and nearly half of them were narrow-range species [20]. Such research focuses on forest reclamation, residual peatlands and polluted land. The restoration capacity of spontaneous plants in these areas has been evaluated. In addition, the factors affecting the growth of spontaneous plants and their ecological benefits have been discussed [21–26]. Some scholars have suggested that spontaneous plants can be applied to newly built or rebuilt road edges, using the repair ability of native plants to replace technical reclamation [27]. In a survey of public perception, it was found that the higher the urbanization level of residential areas, the greater the residents' preference for a natural landscape [28]. Although the public's acceptance of native plants is increasing, there is a threshold effect [29]. Professionals working with plants will pay more attention to spontaneous plants and perceive higher plant richness than the general population [30]. In short, the ecological value of spontaneous plants has become increasingly prominent, and their ornamental value has gradually been recognized.

Wetlands are one of the ecosystems with the richest biodiversity in nature. They play an important role in urban ecosystem management and flood control. However, the rapid development of cities has brought great pressure to and poses a threat to urban wetlands [31]. This is especially true in developing countries [32–34]. There is a large number of spontaneous plants in wetlands, and they play an important role in these ecosystems. Therefore, the construction of sustainable wetland ecosystems is inseparable from the rational utilization of spontaneous plants. To fully understand the key factors affecting the growth and distribution of wetland spontaneous plants, we analyzed (1) the diversity and distribution pattern of spontaneous plants in wetland habitats; (2) the effect of biological characteristics on the composition and distribution of spontaneous plants; and (3) the influence of the surrounding environment on the composition and distribution of spontaneous plants.

## 2. Materials and Methods

### 2.1. Study Area

Hangzhou city, Zhejiang Province, is located on the southern edge of the Yangtze River Delta and Qiantang River Basin (30°12′ N, 120°8′ E). Hangzhou has a subtropical monsoon climate with four distinct seasons and abundant rainfall. The mean annual temperature of the city is 15.9–17 °C, and the mean annual precipitation is approximately 1100–1600 mm. Jiangyangfan Ecological Park (hereinafter referred to as Jiangyangfan Park) is located in the southeastern part of the West Lake Scenic Area. We chose Jiangyangfan Park as our research site, which is in a valley surrounded by mountains on three sides, covering a total area of 19.8 hm² (Figure 1). With the development of ecological thought, Jiangyangfan Park became the first ecological park in China based on the "wild" concept [35]. The park could be roughly divided into three habitat types [36]: artificial and secondary forest, forest and wetland, with slightly alkaline soil [37]. The park used to be a silt reservoir for the dredging of West Lake. As the surface silt dries naturally, it gradually changes from lake to swamps. It has now become a secondary wetland. Since development and construction, park wetland plants have experienced more than 20 years of natural succession, forming a wide range of natural plant communities. Due to surface subsidence over time, hygrophytes such as *Salix rosthornii* have been affected by flooding stress for a long time [38]. The number of hygrophytes, such as *Phragmites australis* and *Hydrocotyle verticillata*, has increased yearly [39].

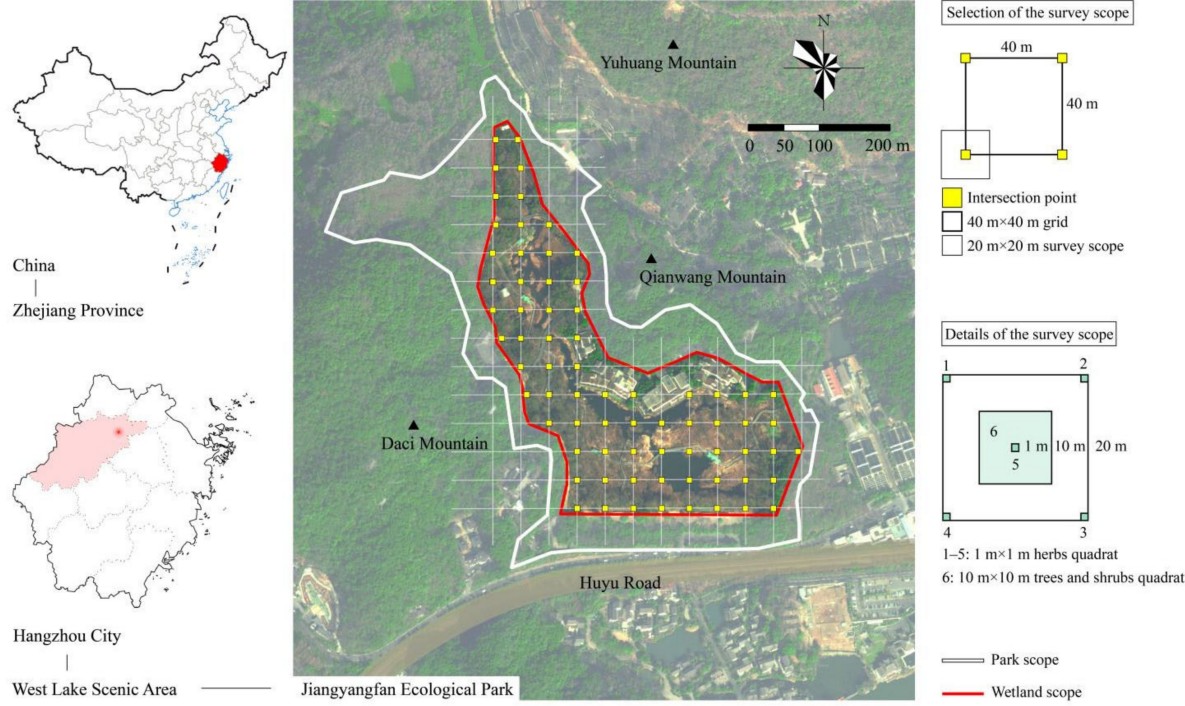

**Figure 1.** Study site and sampling design.

### 2.2. Plant Data

In this study, we investigated spontaneous plants and environmental factors in the wetland of Jiangyangfan Park from 1st April to 13th April in 2022. A number of 40 m × 40 m grids were arranged in the wetland. After adjusting the grid, the maximum number of valid intersections was 71. The selection of some intersections was fine-tuned according to the actual situation (Figure 1). The geographic coordinates of the intersection points were recorded by a Global Positioning System (GPS). Taking the intersection as the center, we set a sample plot range of 20 m × 20 m. We set 1 m × 1 m herbs quadrats at the corners and center of the sample plot. A 10 m × 10 m trees and shrubs quadrat was set in the center.

After excluding the sampling points with no plants, such as water areas, buildings, roads and other out-of-bound sampling points, 334 quadrats were retained. There were 279 herbs quadrats and 55 trees and shrubs quadrats. We took landscape elements and functions as the first-class classification index. Habitat was divided into four types: roadside vegetation area, ecological restoration area, habitat island and drainage area (Table 1). Then, according to human factors and matrix as the secondary classification indicators, the habitat was further divided into six microhabitat types (Figure 2).

**Table 1.** Division and definition of habitats and microhabitats.

| Habitat Type | Microhabitat Type | Feature Description | Artificial Interference Intensity | Number of Quadrats |
|---|---|---|---|---|
| Roadside vegetation area | Cultivated vegetation | Artificially cultivated ornamental vegetation ≤5 m from roadside | High intensity, artificial planting to create seasonal landscape | 24 |
| | Semi-cultivated vegetation | Semiartificial vegetation ≤5 m from roadside | Medium intensity, appropriate artificial planting | 42 |
| Ecological restoration area | Swamp | Long-term overwet land under the surface, with water depth between 0 m and 0.5 m | Low intensity, unattended area | 183 |
| | Secondary forest | Woodland mainly covered by spontaneous plant communities | Low intensity, unattended area | 16 |
| Habitat island | – | Area enclosed by weather-resistant steel | Low intensity, unattended area | 20 |
| Drainage area | Shallow-water area | Water with depth between 0.5 m and 1 m | Medium intensity, appropriate artificial planting | 22 |
| | Deep-water area | Water with depth >1 m | Medium intensity, proper water quality maintenance | 27 |
| Total | | | | 334 |

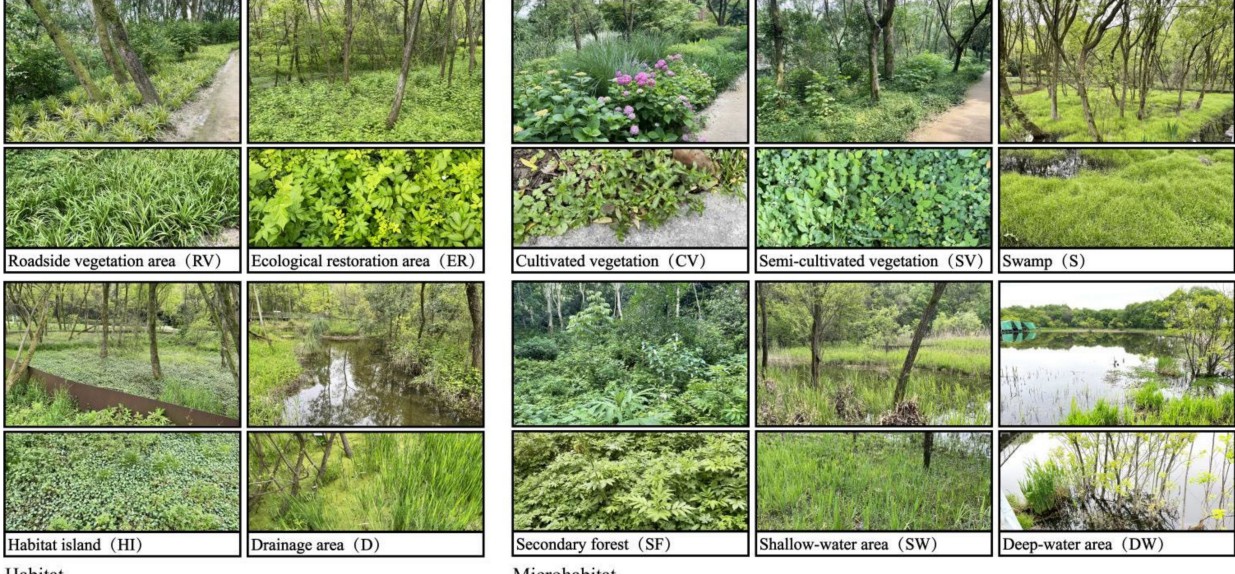

Roadside vegetation area （RV）　Ecological restoration area （ER）　Cultivated vegetation （CV）　Semi-cultivated vegetation （SV）　Swamp （S）

Habitat island （HI）　Drainage area （D）　Secondary forest （SF）　Shallow-water area （SW）　Deep-water area （DW）

Habitat　　　　　　　　　　　　Microhabitat

**Figure 2.** Present status of four habitats and six microhabitats.

The investigated spontaneous plants were photographed by Sony DSC-RX10M4 (with telephoto lens) and iPhone 12. Common plants were identified by the authors themselves, and occasional plants were identified by experts. This study determined spontaneous plant species based on the Flora of China (http://www.iplant.cn/, accessed on 5 October 2022) and divided the fruits of spontaneous plants into eighteen types. Whether or not they were a native species was judged according to the Flora of Hangzhou (http://db.hzbg.cn/hzflora/flora/, accessed on 18 August 2022). According to the Ministry of Ecology and Environment of the People's Republic of China (https://www.mee.gov.cn/, accessed on 18 August 2022), the published list of invasive alien plants in China, and related studies such as Yan et al. [40] and Zhang et al. [41], the spontaneous plants were defined as invasive and non-invasive plants. According to the seed database of the Royal Botanic Garden, Kew, UK (http://data.kew.org/sid/. accessed on 18 August 2022) and related literature [42–44], the seed dispersal modes were divided into six types: animal dispersal, wind dispersal, hydraulic dispersal, autologous dispersal, mixed dispersal and uncertain.

*2.3. Environmental Data*

We selected biological factors and abiotic factors as two major environmental factors for investigation. Biological factors included the growth index and aggregation degree of related cultivated plants in the sample with a growth period of more than two years. Abiotic factors included ecological factors and topographical factors. Ecological factors included soil temperature, soil moisture, soil pH, water pH, water depth and illuminance (Lux). Topographical factors included slope and the direction of slope (slope aspect). The environmental factor investigation was conducted from April 8th to April 10th after eight consecutive days without rain. It was sunny for three days of investigation. A professional soil moisture tester, soil thermometer and soil pH tester (pH328, Xima, Guangdong, China) were used to record the soil moisture, temperature and pH value, respectively. A water quality tester (pH-100B, Lichen, Guangdong, China) was used to measured water pH; repeat measurements were taken by washing with pure water after each use. The water depth was measured by visual inspection. A professional-grade lux meter (TES-1332A, Taishi, Taiwan, China) was used to measure illuminance. An inclinometer was used to measure the slope, and a mobile phone function was used to record the direction of the slope. The measurements of all environmental factors were repeated three times in each sample and recorded. The average values were taken for subsequent analysis.

*2.4. Statistical Analysis*

(1) Data collation and calculations were carried out in Excel and R (v. 4.2.0). The circlize package in R [45] was used for visualization of spontaneous plant patterns in different habitats. The vegan package [46] and the picante package [47] were used to perform diversity analysis of various points. Because all the data had non-normal distributions, nonparametric analysis methods (Kruskal-Wallis test) were used to calculate the data differences.

(2) When calculating the important value (IV), based on the existing literature [48], we also considered the particularity of the site. The calculation of importance values of trees is different from that of herbs and shrubs.

$$S_i(\text{trees}) = \frac{(RA_i + RF_i + RC_i)}{3} \tag{1}$$

$$S_i(\text{herbs and shrubs}) = \frac{(RA_i + RF_i + RD_i)}{3} \tag{2}$$

where $S_i$ is the importance value for the $i$th species, and $RA_i$, $RF_i$, $RC_i$ and $RD_i$ represent the relative abundance, frequency, coverage and dominance, respectively, for the $i$th species.

Species with less than 1% importance value and less than 3% occurrence frequency were eliminated from the distribution pattern. Finally, 24 dominant spontaneous plants were obtained. The population distribution pattern was determined based on the pop-

ulation diversity (number of individuals) as the population quantity characteristic. We used the K-means clustering method to classify plant communities. According to the importance of species, we selected species whose frequency of occurrence was ≥3% at two habitat scales. In the within curve, there was a gradual smoothing trend when dividing the environment into three types of communities. Therefore, each habitat was divided into three community groups. Then, we compared the sum of importance values of the tree, shrub and herb layers to obtain the dominant layer. We named the dominant species with the greatest importance values in each plant community layer according to the biological community.

(3) The TRASP slope direction calculation formula was applied in the processing of environmental factor data [49]:

$$\text{TRASP} = \frac{1-\text{COS}\left((\pi/180)\left(\text{aspect}-30\right)\right)}{2}$$

The slope data are converted into values between 0 and 1. The larger the converted values are, the drier and hotter the habitat conditions. The smaller the values are, the wetter and colder the habitat conditions.

(4) Because the data (environmental factors and species) did not obey a normal distribution, a nonparametric test (Kruskal-Wallis rank sum test) was adopted for comparisons between groups. When there was a significant difference between the groups, the Dunn test or Wilcoxon test was used for multiple comparisons. The linkET package [50] was used to display the Spearman correlation coefficient between environmental factors. In this process, the Bray-Curtis distance was calculated for species, and the Euclidean distance was calculated for environmental factors. Then, the Mantel correlation between species characteristics and environmental variables was calculated. Redundancy analysis (RDA) was performed in the vegan package. After Hellinger transformation, multiple regression analysis was carried out on the response variables of the environmental factors. The rdacca.hp package [51] was used to run variation decomposition and hierarchical segmentation after RDA to obtain the contribution of the explanatory variables of the environmental factors.

## 3. Results

### 3.1. Composition of Spontaneous Plant Species

A total of 100 species of spontaneous plants belonging to 93 genera and 48 families were recorded. Compositae, Rosaceae and Gramineae were the top three families in terms of plant species, with 10, 8 and 7 species, respectively. There were 78 native plants, accounting for 78.00%. There were 22 alien plants, which could be divided into 11 exotic species and 11 exotic invasive species. Invasive plants were mainly from North America, Europe and Africa. *Erigeron annuus*, originally an invasive plant, has become a naturalized species. Spontaneous plants were composed of nine life forms. Of these spontaneous plants, 45 (45%) were perennial herbs, belonging to 40 genera and 23 families. There were 16 species of annual herbs. Semi-shrub plants only contained one species, *Sambucus javanica*. In terms of frequency, the plants with the highest frequency were *Hydrocotyle verticillata* (8.91%), *Phragmites australis* (8.32%) and *Lemna minor* (7.23%). The abovementioned hygrophytes grew well in wetland environments and were widely distributed in various habitats.

Fruits were divided into eighteen types, including achene (19), capsule (19), meristem (16), drupe (7) and caryopsis (7). In terms of seed dispersal, species of animal-dispersed plants were the most abundant, with 35 species belonging to 34 genera and 21 families. Rosaceae was the most abundant family, with 7 species. There were 22 species of wind-dispersed plants belonging to 20 genera and 13 families. Most of the wind-dispersed plants were Compositae, accounting for 36.36%. There were sixteen types of mixed dispersal and fourteen types of uncertain dispersal modes. The reason for uncertainty was that no relevant literature or report explicitly pointed out the specific dispersal mode. The invasive plants were mainly distributed in the mixed dispersal mode (4).

### 3.2. Diversity and Distribution Pattern of Spontaneous Plants in Different Habitats

There were 77 species belonging to 71 genera and 40 families found in ER, which had most species among the four habitat types. Among them, there were 61 spontaneous plants, accounting for 79.22%. Only 15 species were found in ponds. RV, HI and ER contained all plant life forms. There were only three types of life forms in D, annual herbs, perennial herbs and trees. In microhabitats, the spontaneous plants distributed in S were the most abundant (67 species). *Oenanthe javanica* was found in six microhabitats. It has a strong reproductive capacity. *Galium spurium* prefers a humid environment and grew most intensively in S. *Salix magnifica* grew best in SF. *Salix rosthornii* had the strongest adaptability in SW (Figure 3).

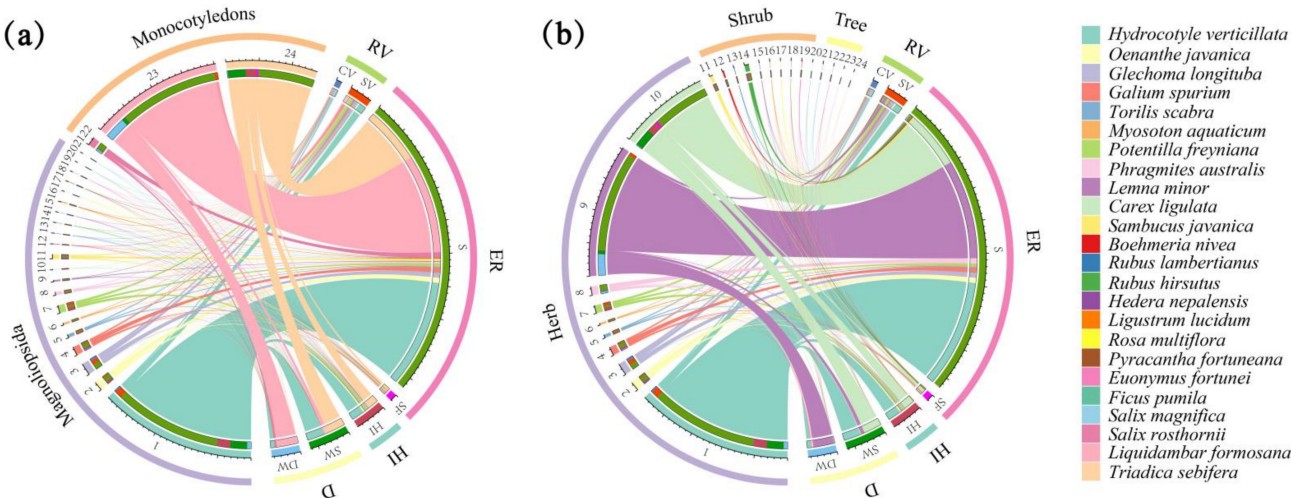

**Figure 3.** (**a**) Chord diagram showing the relationship between the genera of dominant spontaneous plants, the habitats at two scales and the number of species; (**b**) chord diagram showing the relationship between the biological characteristics of dominant spontaneous plants, the habitats at two scales and the number of species.

Diversity analysis showed that there were significant differences in species diversity and evenness of spontaneous plants between the two habitat scales ($p < 0.01$) (Figure 4). In terms of dominant layer and dominant species, the community structure was divided into three types at the habitat scale. These were the tree layer, shrub layer and herb layer. *Salix magnifica* was the dominant species in the ER and HI. Under the influence of water level stress, *Salix magnifica* and *Salix rosthornii* became independent communities in HI and D, respectively. At the microhabitat scale, the dominant layer of herbaceous plants accounted for 50%, which appeared nine times. This was followed by the tree layer, which appeared six times. Herbs with higher dominance were distributed in clusters. There were three modes of shrub distribution: aggregate distribution, uniform distribution and random distribution. *Salix magnifica*, the dominant tree, was uniformly distributed in all habitats. After the effect size (es) analysis, it was found that the diffusion coefficient, aggregation index, average crowding degree and green index had great significance in different habitat scales (es $\geq$ 0.14). Compared with the managed urban green space, the different habitat conditions in the wetland made the distribution of spontaneous plants significantly different.

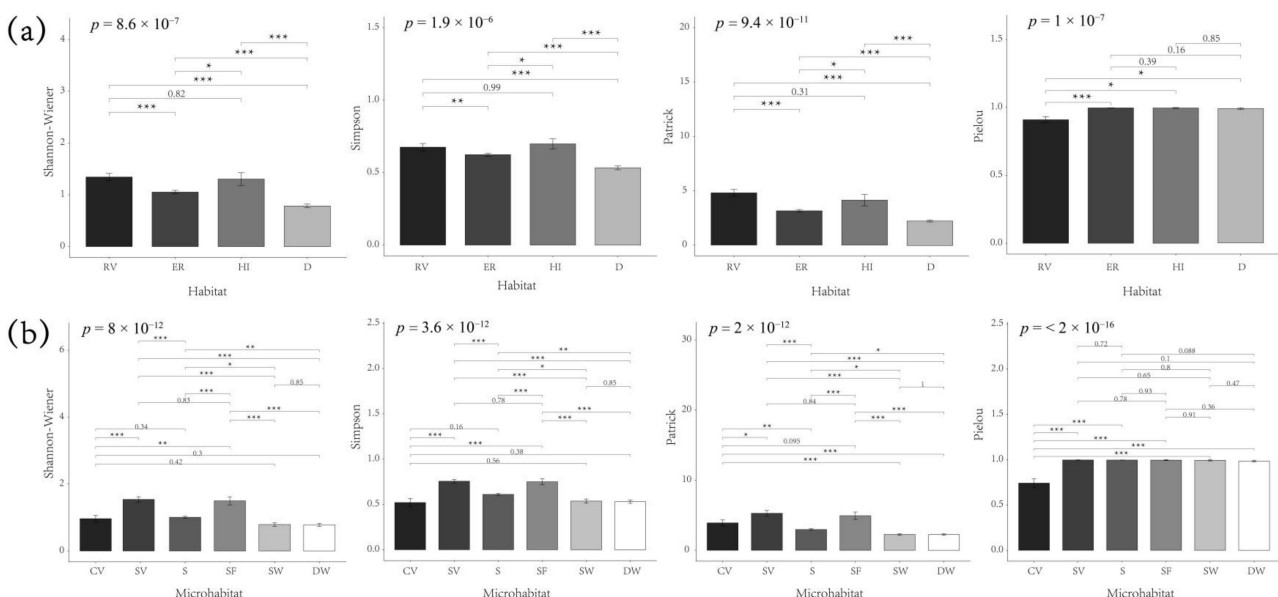

**Figure 4.** Diversity and evenness index of each habitat (**a**) and microhabitat (**b**). Note: *** $p < 0.001$; ** $p < 0.01$; * $p < 0.05$.

### 3.3. Fruit Types and Seed Dispersal of Spontaneous Plants at Different Habitat Scales

At the habitat scale, RV had the richest fruit types, with seventeen types. The most common ones were capsule (17), achene (14) and schizocarp (9). This was because at the habitat scale, RV had high plant species richness and biodiversity. This was followed by ER (16) and HI (13). At the microhabitat scale, SV had the most fruit types. There were sixteen types, including capsule (16), achene (13), schizocarp (8) and caryopsis (6). The types of fruits in DW (6) and SW (5) were relatively singular. The number of fruit types was often positively related to the diversity of spontaneous plants at both the habitat and microhabitat scales.

At the habitat scale, every habitat type contained all the seed dispersal modes. The most animal-dispersed species were found in ER, namely, 27 species belonging to 26 genera and 16 families, among which there were as many as seven species of Rosaceae. Invasive plants were mainly concentrated in RV (10) and ER (7). *Iris pseudacorus* and *Cyperus involucratus* were intentionally introduced as ornamental plants. No water-dispersed plants were found in the three microhabitats (CV, SV and SF) without water flow. However, SV and SW had the most animal-dispersed plants (23) and self-dispersed plants (7). Invasive plants were mainly comprised of semi-cultivated plants (8) and cultivated plants (7), with mixed dispersal plants as the main species. *Setaria palmifolia* is a specie with ornamental value that is planted in a band along the roadside and propagated through mixed dispersal.

### 3.4. Effects of Different Environmental Factors on the Distribution of Spontaneous Plants

Among the environmental factors, the diversity index of cultivated plants (cultivated plant Simpson) had the strongest positive correlation with its clustering degree (Spearman's r = 0.60). Additionally, there was a strong negative correlation between water depth and aspect (Spearman's r = −0.45). There was no correlation between soil pH and other environmental factors. In the Spearman correlation between spontaneous plants and environmental factors, the explanation rate of Mantel's r was not high because of the large incidental components. However, we could still obtain some information from the analysis. There was a strong correlation between the spontaneous plant cover and the clustering degree of cultivated plants ($p < 0.01$). Environmental factors, such as soil temperature, soil moisture and slope, have a large impact on the characteristic index values of spontaneous plants ($p < 0.01$) (Figure 5).

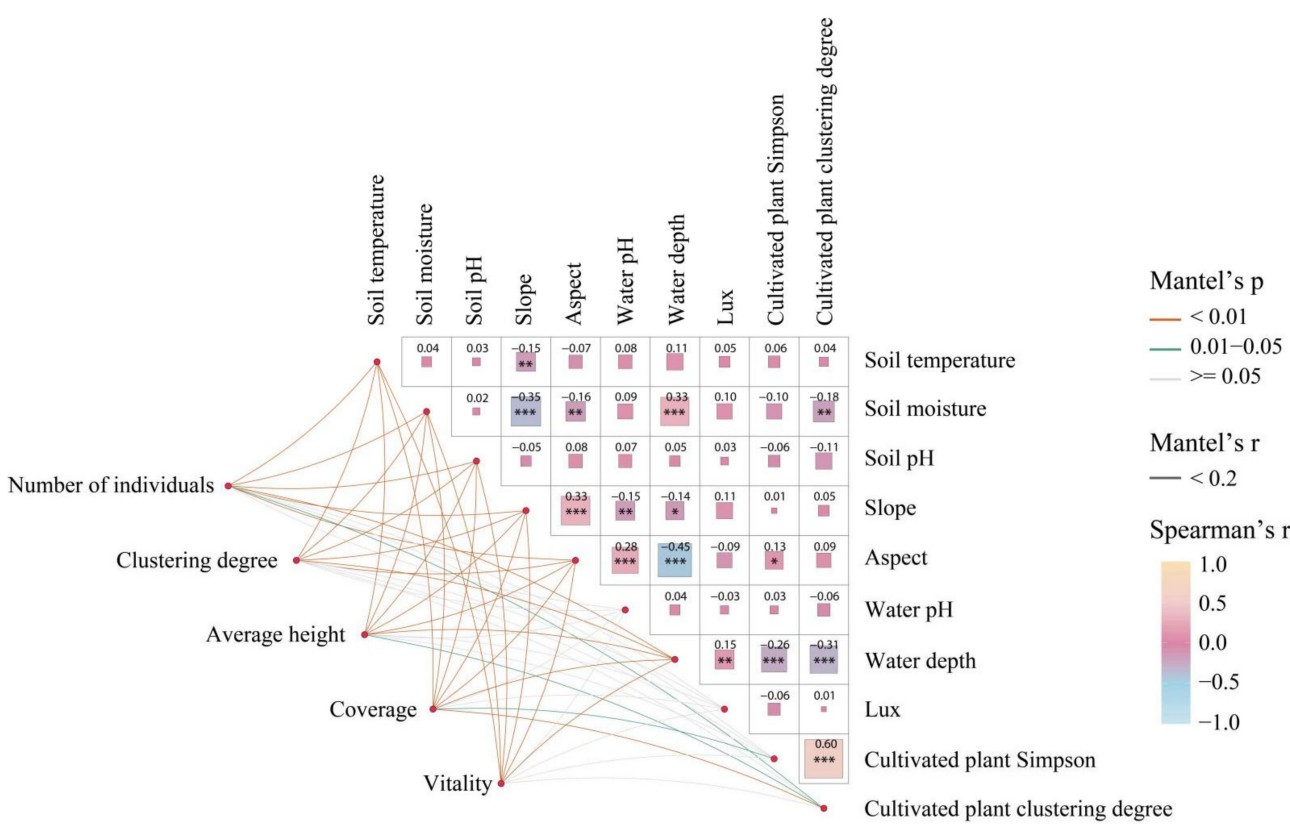

**Figure 5.** Mantel correlation coefficient between dominant spontaneous plants community and various environmental factors, and significance. Note: *** $p < 0.001$; ** $p < 0.01$; * $p < 0.05$.

When analyzing the influence of environmental factors on species composition, it was found that plants such as *Ligustrum lucidum*, *Pyracantha fortuneana*, *Salix rosthornii* and *Rubus lambertianus* were suitable for growing in dark and humid environments on the habitat scale. *Glechoma longituba*, *Torilis scabra*, *Potentilla freyniana* and other plants were greatly affected by topography. They are not suitable for strong alkaline and humid environments and mainly grow in ER. As aquatic plants, *Phragmites australis* and *Lemna minor* showed strong hydrophilicity and adapted to alkaline water quality. The corresponding environmental conditions of *Hydrocotyle verticillata* and *Carex ligulata* included moist soil and gentle terrain (Figure 6a). The diversity index and aggregation degree of cultivated plants caused some interference with plant species composition. The corrected $R^2$ value was small at 3.8%. The water depth had the highest interpretation rate, accounting for 77.11%. This was followed by the aspect, accounting for 25.53% (Table 2). At the microhabitat scale, more than ten types of spontaneous plants, such as *Rubus lambertianus*, *Hedera nepalensis*, *Salix rosthornii* and *Pyracantha fortuneana*, all showed warm-loving growth characteristics, and most of them grew in SV. The distribution of swamp samples was concentrated in the first quadrant. Among them, the spontaneous plants were positively correlated with water conditions and water depth and were most affected by water level (Figure 6b). The $R^2$ value of the corrected microhabitat was slightly higher than that of the habitat, which was 4.2%. Water depth and aspect were still the top two environmental factors in terms of the interpretation rate, accounting for 67.86% and 23.81%, respectively (Table 2).

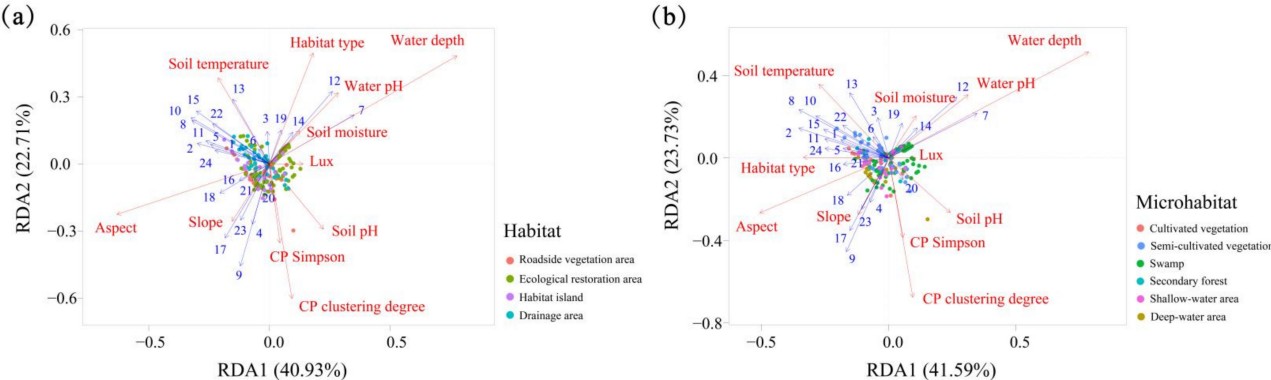

**Figure 6.** RDA ordination diagram of 24 dominant species and environmental factors at habitat (**a**) and microhabitat (**b**). 1: *Ficus pumila*; 2: *Hedera nepalensis*; 3: *Salix magnifica*; 4: *Myosoton aquaticum*; 5: *Liquidambar formosana*; 6: *Euonymus fortunei*; 7: *Lemna minor*; 8: *Rubus lambertianus*; 9: *Glechoma longituba*; 10: *Pyracantha fortuneana*; 11: *Sambucus javanica*; 12: *Phragmites australis*; 13: *Salix rosthornii*; 14: *Hydrocotyle verticillata*; 15: *Ligustrum lucidum*; 16: *Rubus hirsutus*; 17: *Torilis scabra*; 18: *Potentilla freyniana*; 19: *Carex ligulata*; 20: *Oenanthe javanica*; 21: *Triadica sebifera*; 22: *Rosa multiflora*; 23: *Galium spurium*; 24: *Boehmeria nivea*; CP Simpson: cultivated plant Simpson index; CP clustering degree: cultivated plant clustering degree.

**Table 2.** Hierarchical segmentation results of environmental factors at the habitat and microhabitat scales.

| Environmental Factor | Habitat | | | | Microhabitat | | | |
|---|---|---|---|---|---|---|---|---|
| | Unique [1] | Average Share [2] | Individual [3] | I Perc (%) [4] | Unique | Average Share | Individual | I Perc (%) |
| Soil temperature | $-0.0025$ | $5 \times 10^{-4}$ | $-0.002$ | $-5.26$ | $-0.0028$ | $3 \times 10^{-4}$ | $-0.0025$ | $-5.95$ |
| Soil moisture | $9 \times 10^{-4}$ | $7 \times 10^{-4}$ | $0.0016$ | $4.21$ | $0.0018$ | $5 \times 10^{-4}$ | $0.0023$ | $5.48$ |
| Soil pH | $-0.0029$ | $-1 \times 10^{-4}$ | $-0.003$ | $-7.89$ | $-0.0026$ | $1 \times 10^{-4}$ | $-0.0025$ | $-5.95$ |
| Slope | $-0.002$ | $8 \times 10^{-4}$ | $-0.0012$ | $-3.16$ | $-0.0028$ | $9 \times 10^{-4}$ | $-0.0019$ | $-4.52$ |
| Aspect | $0.0031$ | $0.0066$ | $0.0097$ | $25.53$ | $-0.001$ | $0.011$ | $0.01$ | $23.81$ |
| Water pH | $0.0012$ | $0.0015$ | $0.0027$ | $7.11$ | $7 \times 10^{-4}$ | $0.002$ | $0.0027$ | $6.43$ |
| Water depth | $0.0209$ | $0.0084$ | $0.0293$ | $77.11$ | $0.0136$ | $0.0149$ | $0.0285$ | $67.86$ |
| Lux | $-0.0013$ | $-5 \times 10^{-4}$ | $-0.0018$ | $-4.74$ | $-7 \times 10^{-4}$ | $-8 \times 10^{-4}$ | $-0.0015$ | $-3.57$ |
| Habitat type | $0.0016$ | $0.0023$ | $0.0039$ | $10.26$ | $0.0039$ | $0.0028$ | $0.0067$ | $15.95$ |
| CP Simpson | $-0.0029$ | $0.0012$ | $-0.0017$ | $-4.47$ | $-0.003$ | $0.0013$ | $-0.0017$ | $-4.05$ |
| CP clustering degree | $-8 \times 10^{-4}$ | $0.0014$ | $6 \times 10^{-4}$ | $1.58$ | $0$ | $0.0019$ | $0.0019$ | $4.52$ |

[1] Unique: the proportion of the total variation explained separately by each explanatory variable. [2] Average share: the division of the common explanation part of each explanatory variable and other explanatory variables. [3] Individual: the proportion of each explanatory variable to the total variation. [4] I perc (%): the proportion of each explanatory variable to the total explained variation; bold numbers have the highest rate of explanation.

## 4. Discussion

### 4.1. Habitat Heterogeneity of Spontaneous Plants

A total of 100 species of spontaneous plants was found in the wetland area of the studied park. Comparative studies with other municipalities directly under the central government and provincial capital cities in China revealed that the number of spontaneous plants was similar to that in Xi'an (provincial capital city, 95 species) [52]. However, there was a significant difference from the survey results of indigenous plants in the mountain

cities of Chongqing (municipality directly under the central government, 279 species) [20] and Kunming (provincial capital city, 386 species) [53]. It is possible that, compared with cities mainly composed of plains, such as Hangzhou and Xi'an, Chongqing (mainly composed of hills and mountainous regions with complex and variable topography) and Kunming (a low-latitude plateau with a large topographic elevation difference) both had distinct regional characteristics. When urban growth was at the same level, topography had a positive correlation with the diversity pattern. This was one of the most important factors affecting plant diversity [20]. Compositae was the family with the largest proportion in most of the studies of spontaneous plants [54,55]. This was consistent with the results of this survey. This is because Compositae, as the largest family of angiosperms, contains more plant species. It is worth mentioning that Compositae accounts for a large proportion of alien species in China and does great harm [56,57]. So, these plants are very adaptable to the environment and have a strong ability to spread. They have the potential advantage of being spontaneous plants. In the wetland of Jiangyangfan Park, native plants accounted for 78% of the total. Among the alien invasive plants, perennial herbs accounted for 55.56%. It has been found that the proportion of annual invasive plants in China has increased significantly, while that of perennial plants has decreased significantly [58]. The overall status of spontaneous plants in Jiangyangfan Park was different from the above research results. This may be because perennials such as *Hydrocotyle verticillata*, *Setaria palmifolia* and *Cyperus involucratus* are often intentionally introduced as ornamental plants [59] in the small-scale habitats of park wetlands. Human factors increased their proportion. It is also possible that perennials grow more tenaciously and survive better in changing environments. It should be noted that *Hydrocotyle verticillata* thrived on a large scale in the wetlands of Jiangyangfan Park. Research has shown that invasive aquatic plants can threaten the stability and biodiversity of wetland ecosystems [60]. In China, *Pontederia cordata*, *Acorus calamus*, *Iris wilsonii* and *Myriophyllum aquaticum*, common wetland species, were used in an experiment to explore the impact of invasion by *Hydrocotyle vulgaris* into these wetland communities. Research confirmed that the diffusion and invasion ability of *Hydrocotyle vulgaris* into aquatic habitats was very low [61]. Some wetland species distribution patterns in Jiangyangfan Park were almost consistent with that of the above communities; thus, the conclusion of the described study has some reference value. Preventive and control measures could be applied to *Hydrocotyle verticillata* in the wetland of Jiangyangfan Park when necessary.

Among the four habitat types, the spontaneous plant species in ER were the most abundant, accounting for 77% of all species. However, from the number of species found at a single sampling point, the calculated value was only 0.42 per point. Through edge expansion and diffusion ability, Cyperaceae naturally colonizes swamp areas [62] and seizes the living space of other spontaneous plants. However, the SF was covered by a large number of shrubs and trees, and spontaneous plants could not obtain adequate growth resources in this high-canopy-density environment, which tended to give priority to climbing plants. Interestingly, in SF, the result was not different from the expectation that the higher the diversity of trees and shrubs, the higher the diversity of native climbers. In reality, there was no connection between the two. The interaction between the arbor-shrub host and the climber is the main reason for climbing plant diversity [63]. However, the role of the specific aspects of each has yet to be elucidated. The diversity index was the lowest in D despite it having appropriate hydrophytes, such as *Pontederia cordata* and *Nymphaea tetragona*, planted for ornamental purposes. The diversity index for D was significantly different from that of other habitats ($p < 0.01$). We consulted park managers and found that *Pontederia cordata* has grown poorly under stress from spontaneous plants in recent years and needs continuous replanting. However, *Nymphaea tetragona* were still growing steadily. A comparative experiment of plant diversity has been carried out in artificially planted and natural succession ponds. The results showed that the diversity of plants in the artificial ponds gradually decreased, whereas the diversity of plants in natural succession ponds first increased and then decreased, and the species composition between the two

was similar after four years [64]. As the wetlands in Jiangyangfan Park were always part of the park, the visual experience of visitors should be considered, making it necessary to plant some ornamental plants. To avoid their excessive encroachment in water, plants such as *Lemna minor*, which spread rapidly and will affect the landscape aesthetics, should be cleared regularly.

Although CV and SV both belonged to the RV habitat, the diversity index of spontaneous plants was significantly different. The largest difference between the two microhabitats lies in the intensity of human interference. The common impression was that human intervention would lead to intensified species homogenization. Recent studies have shown that reasonable human intervention can actually improve biodiversity to some extent [65]. Although the diversity of spontaneous plants was adversely affected by the conservation of cultivated plants, the biodiversity of RV was significantly higher than that of other habitats because of the large number of honey plants planted. Studies have indicated that intervention and reprivatization are not mutually exclusive, but complementary strategies need to be found [66]. The plant management model of Jiangyangfan Park was similar to "rewilding when possible and intervening where needed", which is significant for the construction strategy of harmonizing the conflict between humans and nature.

### 4.2. Effects of Biological Characteristics of Spontaneous Plants on Distribution

The growth of spontaneous plants depended to a great extent on biological and environmental factors. Seed propagation determines the distribution of plants in the process of natural regeneration [67] and affects the population and diversity of plants [68]. In Jiangyangfan Park, achenes and capsules were the most common fruit types. This was consistent with the research results of fruit types of spontaneous plants in Fuzhou roadside ponds [69]. Achenes are small, dry fruits with a hard, noncracking peel and light in weight. According to the survey results, achenes were mostly spread by wind. Wind-dispersed species were mainly concentrated in the family Compositae. This is because the seeds of compound fruits are not only numerous but also have appendages, referred to as setae [44]. This accessory structure aids in wind dispersal. Jiangyangfan Park is located in a valley where the wind speed is high. Plant seeds could naturally be dispersed over a long distance by wind, and they appeared in all types of habitats. Capsule-type fruits have various cracking modes at maturity, and most show autologous seed dispersal, such as in *Myosoton aquaticum* and *Oxalis corniculata*. However, in this study, wind dispersal was still the main mode of capsule dispersal. Wind-dispersed plants were tall trees, such as *Salix rosthornii*, *Salix magnifica* and *Liquidambar formosana*. *Salix rosthornii* and *Salix magnifica* are high-frequency species that appear in various habitats. This not only showed that the two plants had high adaptability to wetland environments but also confirmed that the dispersal distance was related not only to wind speed but also to the height of seed release [70].

In addition to the influence of wind on seed dispersal, animal-dispersed plants accounted for the largest proportion (32%). This was inconsistent with the result in Nanjing Urban Park, where self-propagating plants made up the highest proportion. This may have been due to development and construction, which filled the city with high-rise buildings and made biodiversity relatively low, conditions that limited dispersal by animals and wind [71]. As an ecological park, Jiangyangfan, which focuses on ecological restoration, is rich in biodiversity and plant resources. Among the animal-dispersed plants, most fruit types were drupes, berries and achenes. Drupe and berry are fleshy fruit, and they tend to attract birds, mammals or insects that eat fruit. Some birds have migratory habits, and the direction and distance to which birds spread plant seeds in active dispersal usually change seasonally [72]. There were approximately 40 species of birds in Jiangyangfan Park, and the more common ones were *Passer montanus*, *Paradoxornis webbianus* and *Spizixos semitorques*. Among them, *Spizixos semitorques* preferred to eat the fruits of *Hedera nepalensis* and *Paederia scandens*. Therefore, the plant seeds moved along with it over a large geographical span. Animal digestive tract dispersal is one of the dispersal modes of animals. During the study in Jiangyangfan Park, some vertebrates were found, such as wild cats and skinks, and

fruits were part of their diets. Some plant seeds were scattered around the park by the activities of vertebrates. Human activities are also an important part of plant dispersal. Some plants had strong ornamental values and were intentionally planted, such as *Sapium sebiferum*, *Pterocarya stenoptera* and *Orychophragmus violaceus*. They breed in the park by spreading their own seeds in different ways. Cultivated plants become self-generated plants through secondary growth. *Nelumbo nucifera* was unique regarding the dispersal mode of spontaneous plants. It came to Jiangyangfan Park along with the silt dredged from West Lake, without any human intervention except transportation. Its aggregate fruit was propagated by seeds and rhizomes, and it became the dominant species in some water systems. In addition, *Hydrocotyle verticillata*, *Potentilla angustifolia*, *Glechoma longituba* and so on undergo asexual propagation. Among them, the number of individuals of *Hydrocotyle verticillata* in S accounted for nearly one-third of the total. This species can grow in a large area in a suitable environment and even occupy the living space of other native plants. Invasive plants should be rationally used according to their different dispersal modes to avoid large-scale spread.

### 4.3. Influence of Environmental Factors on the Distribution of Spontaneous Plants

In the analysis of environmental factors, the linkET package was used to analyze the relationship between environmental factors and the biological characteristics of spontaneous plants. The smaller the slope aspect value after conversion, the wetter and colder the habitat conditions. Therefore, slope aspect had a strong negative correlation with water depth. In addition, the results showed a strong positive correlation between soil moisture and water depth, which was consistent with previous research [73]. In the selection of environmental factors, considering that the relationship between spontaneous plants and cultivated plants was interdependent, some indexes of cultivated plants were intentionally added as environmental variables. Based on the results, the cover of spontaneous plants and the diversity index of cultivated plants both had a relationship with the clustering degree. Between them, the correlation with the clustering degree of cultivated plants was stronger. This may be because the clustering degree was positively correlated with cover, while cover was negatively correlated with illumination. Plants can adjust their own growth and development by sensing changes in light and temperature [74]. However, in this study, when light intensity was taken as an environmental variable, it had the least influence on the composition of spontaneous plants at the two habitat scales. This was not consistent with most research conclusions [75–77]. This may be because most of the selected sampling sites were areas with no canopy shading or short-term shading; thus, there was no considerable difference in the average measured value of illumination. Other reasons should be explored.

Through the RDA of species composition and environmental factors, it was found that water depth, slope aspect and cultivated plant clustering degree had a strong influence on the composition of spontaneous plants at the two habitat scales. It should be noted that compared with the explanatory rates of various environmental factors at the habitat and microhabitat scales, the effect of cultivated plant clustering degree on spontaneous plants at the microhabitat scale was significantly greater than at the habitat scale. This indicates that spontaneous plants were very sensitive to change at the fine scale. As the largest influencing factor, water depth contributed 77.11% to the environmental interpretation at the habitat scale and 67.86% at the microhabitat scale. Some scholars specifically listed the suitable density of *Phragmites australis* for planting at different water levels [73], and their sensitivity to changes in groundwater level could be captured by recording information such as traits. With increasing water depth, *Phragmites australis* gradually inhibited the growth of some wetland plants, which also explained the low diversity index of spontaneous plants in D [78]. In this study, the effect of soil pH on the species composition of spontaneous plants was inconsistent with the result that pH in the peat bogs of Latvia was the main driver of spontaneous plants [24]. This may be because the soil in Jiangyangfan Park came from mud dredged from West Lake. There was no significant difference in soil pH among the

different habitats, and the composition of spontaneous plants lacked strong support. It may be because compared with other special studies [79], the chemical elements in soil were not selected as variables in this study. A large number of studies have focused on the ecological restoration ability of spontaneous vegetation in contaminated or damaged land [21,22]. The effect of natural regrowth was more valuable and cost-effective than the artificial allocation mode of forest reclamation [23]. It could be speculated that after 25 years (1997–2022) of natural succession, the saline-alkali soil that was originally not suitable for growth in Jiangyangfan Park had begun to undergo new changes. Of course, more studies are needed to confirm this speculation.

## 5. Conclusions

Hangzhou Jiangyangfan Ecological Park has constructed a plant landscape based on a wild environment, and its design method of no interference is the first case in China. After more than 20 years of natural succession, there are abundant spontaneous plant resources in the wetland, most of which are native plants. Although some invasive plants were introduced intentionally, they did not expand without any interference. The native plants that had grown for many years had already occupied suitable niches. Among the most common plants, *Hydrocotyle verticillata*, as an invasive alien species, needs to be considered due to its negative effects. Achenes and capsules are the main fruit types of the spontaneous plants in the wetland. Animal-dispersed plants are widely distributed in ecological parks due to their rich biodiversity. There was habitat heterogeneity at different scales in the park. In this study, water depth and slope aspect were the key factors that determined plant diversity and distribution patterns. In addition, the clustering degree of cultivated plants also had a great influence on the composition of spontaneous plants. According to the characteristics and sources of plants, we can make rational use of the environmental preferences of spontaneous plants and provide a useful reference for the construction of urban wetland landscapes with low maintenance, high biodiversity and sustainable development.

**Author Contributions:** All the authors of this manuscript have contributed substantially to the work here reported. Conceptualization, Y.Y., B.X., F.S. and X.L.; data curation, Y.Y. and B.X.; writing—original draft, Y.Y. and B.X.; investigation, Y.Y., B.X., Q.Y., L.F., T.G., D.F. and H.C.; visualization, Y.Y., B.X. and H.Y.; resources, F.S. and X.L.; funding acquisition, F.S.; supervision, F.S. and X.L.; project administration, F.S. and X.L.; validation, X.L. All authors have read and agreed to the published version of the manuscript.

**Funding:** This research was supported by the National Key R&D Program of China (NO: 2022YFF1303102), the Project of National Natural Science Foundation of China (NO: 51978627) and the Scientific Research and Training Program of Undergraduates at the Zhejiang Agriculture and Forestry University in China (NO: S202210341012).

**Data Availability Statement:** Not applicable.

**Acknowledgments:** We thank Wenhao Hu and Xiaolu Li for their valuable suggestions and comments on the manuscript. We also express our gratitude to Hangzhou Botanical Garden, for their assistance with this paper.

**Conflicts of Interest:** The authors declare no conflict of interest.

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
