# Peer review of "Distribution Pattern and Factors Influencing Spontaneous Plant Diversity in Different Wetland Habitats"

_forests, doi:10.3390/f13101678_

Round 1

Reviewer 1 Report

The research article entitled ‘Distribution pattern and factors influencing spontaneous plant 2 diversity in different wetland habitats’ is a good piece of work carried out by Yang et al. The manuscript is well written with minor typological/grammatical errors. Above all, the analysis of the data has been efficiently carried out and very well represented. Congratulation for that! I read the ms thoroughly, it was quite interesting. However, I have some suggestion, incorporation of which will definitely make the ms more precise, clear and easy to understand.

1.      Introduction needs more justification; further some more literature should be added from other parts of the globe.

2.      Line 118-120: A total of 40 m × 40 m grids were arranged in the wetland. To include the most effective quadrats, 71  grid intersections were ultimately determined for the investigation of spontaneous plants (any reference that was followed)

3.      Line 124: There were 279 herbaceous quadrats and 55 shrub quadrats. Size of the quadrats should be mentioned.

4.      In Fig. 1; 6 represents trees and shrubs; however there is no mention of trees in text (Line 124)

5.      Line 171; how the importance value was calculated?

6.      There is no mention how the plant species were collected, identified and have the authors deposited the herbariums of plant specimens.

7.      Authors have mentioned that how they classified species into invasive and non-invasive; it should be better to include how the native species were identified.

8.      Which system of classification was followed?

9.      Have the authors collected trees also? If no, then why to mention.

10.  Line 346-348; references should be added.

Author Response

October 8, 2022

Dear Editor and Reviewers,

We greatly appreciate your critical review of our manuscript (forests-1953587). Thank you for your helpful comments and constructive suggestions, which inspired us to perform further analyses of certain problems to enrich this paper.

We have revised the manuscript according to the reviewers’ comments. The attached file included the response to reviewers’ comments, a copy of the revised version with the changes marked and a copy of the revised version with no changes marked.

With kind regards,

Associate Professor Feng Shao

Email address: shaofeng@zafu.edu.cn

Zhejiang Agriculture and Forestry University

Reviewer 2 Report

The diversity and its environmental correlates of spontaneous plants in urban ecosystem is an interesting topic. In the current study, the authors investigated the spontaneous plants of a 20+ year old ecological park in Hangzhou city, and obtained some meaningful conclusions. This paper is well organized, but I think the English of the full text needs to be improved. Here, I just proposed some questions before it was published.

Lines 50-52: I think the authors here could cite a reference here. Xu et al., 2019, Human activities have opposing effects on distributions of narrow-ranged and widespread plant species in China. PNAS, 116(52): 26674-26681.

Line 56: “This was .... What does this refer to? Please rewrite this sentence.

Lines 60-80: I think this paragraph is a bit disorganized. Maybe it can be divided into two paragraphs. In the first paragraph, authors give us the introduction of spontaneous plants”, from “weeds” to “spontaneous vegetation”, and to “spontaneous plants”. So here, the first sentence “Spontaneous plants often grow in urban wastelands” should move to other places followed the introduction of spontaneous plants”. In the second paragraph, authors could summarize some studies about “spontaneous plants” in some cities in China.

Lines 81-82: This sentence should move to a suitable place in the first paragraph of Introduction section.

Lines 89-90: There is no clear background of “habitat” and “microhabitat”. I think the authors could delete this sentence, or add more usefull information.

Lines 113-115: Salix rosthornii, Phragmites australis, and Hydrocotyle vulgaris are species instead of vegetation.

Line 118 & Figure 1: A total of 40 m × 40 m grids ... . How many grids were arranged in the wetland? But the grids in Figure 1 is 20 m × 20 m, is there any different?

Table 1: Number of samples should be Number of quadrats.

Lines 155-164: The (1) section of statistical analysis should move to 2.2 Plant data.

Lines 210: Rewrite this sentence. life form was “perennial herbs”, which were. This is not appropriate.

Line 228: ER had the most species ..... I think should be change into There were 77 species belonging to 71 genera and 40 families found in ER, which harboured most species among the four habitat types.

Line 231: “:” should be change into ,.

Figure 3-6: Please cite these figures in suitable places of the result section.

Lines 334-338: The format of the contents in brackets is not consistent in the four cities.

Line 383: “We asked park managers .... Please put it in another academic way.

Author Response

October 8, 2022

We greatly appreciate your critical review of our manuscript (forests-1953587). Thank you for your helpful comments and constructive suggestions, which inspired us to perform further analyses of certain problems to enrich this paper.

We have revised the manuscript according to the reviewers’comments. The attached file included the response to reviewers’comments, a copy of the revised version with the changes marked, a copy of the revised version with no changes marked and necessary files.

With kind regards,

Associate Professor Feng Shao

Email address: shaofeng@zafu.edu.cn

Zhejiang Agriculture and Forestry University
